# The Role of Obesity in Breast Cancer Pathogenesis

**DOI:** 10.3390/cells12162061

**Published:** 2023-08-14

**Authors:** Ira Glassman, Nghia Le, Aamna Asif, Anabel Goulding, Cheldon Ann Alcantara, Annie Vu, Abraham Chorbajian, Mercedeh Mirhosseini, Manpreet Singh, Vishwanath Venketaraman

**Affiliations:** 1College of Osteopathic Medicine of the Pacific, Western University of Health Sciences, Pomona, CA 91766, USAnghia.le@westernu.edu (N.L.); aamna.asif@westernu.edu (A.A.); cheldonann.alcantara@westernu.edu (C.A.A.); mercedeh.mirhosseini@westernu.edu (M.M.); 2Corona Regional Medical Center, Department of Emergency Medicine, Corona, CA 92882, USA

**Keywords:** obesity, diabetes mellitus, breast cancer, glutathione, mTOR, leptin, BMI

## Abstract

Research has shown that obesity increases the risk for type 2 diabetes mellitus (Type 2 DM) by promoting insulin resistance, increases serum estrogen levels by the upregulation of aromatase, and promotes the release of reactive oxygen species (ROS) by macrophages. Increased circulating glucose has been shown to activate mammalian target of rapamycin (mTOR), a significant signaling pathway in breast cancer pathogenesis. Estrogen plays an instrumental role in estrogen-receptor-positive breast cancers. The role of ROS in breast cancer warrants continued investigation, in relation to both pathogenesis and treatment of breast cancer. We aim to review the role of obesity in breast cancer pathogenesis and novel therapies mediating obesity-associated breast cancer development. We explore the association between body mass index (BMI) and breast cancer incidence and the mechanisms by which oxidative stress modulates breast cancer pathogenesis. We discuss the role of glutathione, a ubiquitous antioxidant, in breast cancer therapy. Lastly, we review breast cancer therapies targeting mTOR signaling, leptin signaling, blood sugar reduction, and novel immunotherapy targets.

## 1. Introduction

According to the American Cancer Society (ACS), an estimated 300,590 new cases and 43,700 deaths due to breast cancer are expected in the US in 2023 [1]. Breast cancer is the most common malignancy diagnosed in women worldwide; it is estimated that the risk of a woman developing breast cancer in her lifetime is 12.9% [1]. Concurrently, the CDC reports that the US is experiencing unprecedented increases in obesity, with obesity prevalence increasing from 30.5% in 2017 to 41.9% in March 2020, and during this period, the prevalence of severe obesity increased from 4.7% to 9.2%.

The growing obesity epidemic has nearly tripled since 1975. Obesity is a complex and multifactorial disease which leads to an increased risk of morbidity and mortality [2]. Obesity can be affected by genetic, behavioral, socioeconomic, and environmental factors. While these factors play an important role, changes in diet and lifestyle have had the biggest impact on the rise in obesity. Globally, there has been an amplified intake of energy-dense foods high in fats and sugar accompanied by a progressively sedentary lifestyle. These changes lead to a difference in energy intake and expenditure that results in obesity. Ultimately, obesity can lead to an increased risk of mortality from cardiovascular disease, diabetes, and cancer [2].

Being overweight increases the risk of developing type 2 diabetes (type 2 DM) by a factor of three and obesity by a factor of seven compared to normal-weight individuals [2]. Type 2 DM results from decreased secretion of insulin by pancreatic beta cells and increased resistance to insulin by insulin-sensitive tissues [3,4,5]. As the sole manufacturing source of insulin, pancreatic beta cells are strictly regulated to ensure that blood glucose concentrations stay within the normal physiological range [6]. However, due to their fragile nature, pancreatic beta cells’ strict regulation is decreased, along with insulin sensitivity, in obese patients [7]. Pancreatic beta cells consequentially become overworked to the point of declination in function, leading to the eventual formation of type 2 DM [5].

Obesity and type 2 DM are associated with increased risk of various cancers, including breast cancer. Additionally, metabolic syndrome, the constellation of visceral adiposity, hyperglycemia, hypertriglyceridemia, and hypertension, is associated with an increase in the risk of breast cancer, breast cancer recurrence, and breast-cancer-specific mortality [8]. Breast cancer can be divided into four major groups based on estrogen receptor (ER), human epidermal growth factor 2 (HER2), Ki-67 index, and progesterone receptor (PR) expression: Luminal A (ER+/HER2-/Ki-67 low/PR+), Luminal B (ER+/HER2+/Ki-67 low/PR low), HER2-enriched (ER low/HER2+/Ki-67 high/PR low), and Triple-Negative Breast Cancer (TNBC, ER-/HER2-/Ki-67 high/PR-) [9]. In a retrospective case-control study by Hossain et al., they found that elevated BMI was a risk factor for Luminal A breast cancer. Elevated BMI and HR- breast cancers yielded no significant relationship. According to their study, there was a significant association between diabetes and three molecular subtypes of breast cancer in Louisiana—Luminal A, TNBC, and Her2+. After controlling for obesity status, the risk was nearly unchanged. These results suggest that diabetes is a strong independent risk factor for breast cancer development [10]. A multiethnic cohort study by Maskarinec et al. found contrasting results when they identified that adjusting for BMI weakens the association of T2DM with breast cancer. They initially found that T2DM increased breast cancer risk by 30%, but after BMI adjustment, the risk fell to only 8% [11]. While both obesity and T2DM are associated with breast cancer development, obesity appears to have a paradoxical relationship conferring protective benefit in premenopausal women and increasing risk in postmenopausal women, whereas T2DM confers risk in both premenopausal and postmenopausal groups. Interestingly, Palmer et al. found that while T2DM increased risk of both ER-positive and ER-negative breast cancer in African American women, a group disproportionately affected by T2DM and ER-negative breast cancer, the hazard ratio (HR) for T2DM and ER-negative breast cancer was highest among nonobese women (1.92, 95% CI, 1.22–3.04) [12].

Understanding the role obesity plays in the pathogenesis of breast cancer is an important source of targeted cancer prevention and development of novel therapeutics to reduce the risk and severity of breast cancer. Obesity and type 2 DM are considered states of chronic inflammation. Research has shown that oxidative stress is both induced by obesity and contributory to its pathogenesis [13]. Reducing reactive oxygen species (ROS) is a point of contention in the treatment of breast cancer as reducing inflammation is viewed as a protective effect; however, ROS are required for the destruction of tumor cells. As such, the role of the ubiquitously present antioxidant, glutathione in its reduced state (GSH), in the treatment of breast cancer requires further investigation.

Body mass index (BMI) is the traditional metric utilized to assign patients into categories of weight and obesity. Unfortunately, BMI is inaccurate in assessing health as it ignores key demographic characteristics that play into weight and height. Research has identified trends in BMI which reflect modifiable breast cancer risk, however. In postmenopausal women, BMI-associated obesity is linked to a higher breast cancer risk, whereas in premenopausal women, BMI-associated obesity is linked to protective effects [14,15,16]. Evidence of BMI’s paradoxical relationship with premenopausal breast cancer pathogenesis varies. A systematic review by Roche et al. found evidence that high BMI in premenopausal women has a tendency towards increased risk for the more aggressive TNBC. They found a decreased risk for less aggressive tumor subtypes such as luminal A. These results suggest that the effects of obesity on breast cancer pathogenesis in premenopausal women may be subtype-dependent [17]. Some studies contrast this and show no significant effect of BMI on breast cancer incidence during the premenopausal period [18]. The differences in results of studies may be attributed to confounding variables rather than a protective effect of BMI on breast cancer incidence. For example, BMI does not consider muscle mass, bone density, overall body composition, and racial and sex differences, which could explain why a premenopausal woman with an elevated BMI, possibly due to increased muscle mass from a healthy lifestyle, would have decreased chances of developing breast cancer.

Hyperglycemia and hyperlipidemia coupled with obesity create a preferential environment for tumorigenesis. Research has demonstrated that cancer cells are highly efficient in competing against neighboring cells for glucose through aerobic glycolysis. Uncontrolled activation of the mammalian target of rapamycin complex 1 and 2 (mTORC1/2) pathway has been implicated in obesity, diabetes, and various cancers, including breast cancer. mTORC1/2 directly modulate glucose utilization, gluconeogenesis, insulin resistance, and they indirectly regulate pancreatic B-cell mass and activity [19]. Several pharmacological agents have been utilized to inhibit the uncontrolled activation of mTORC1 specifically to improve tumor responsiveness to current breast cancer treatments. Leptin, a hormone released from adipocytes, has also been implicated in obesity and breast cancer. Leptin has been shown to increase tumorigenesis and invasion through activation of oncogenic stem cell signaling pathways [20]. Studies have identified leptin, aromatase, insulin receptor, peroxisome-proliferator-activated receptor gamma (PPARγ), and lifestyle changes as other potential targets for obesity-associated breast cancer treatment.

## 2. Obesity and Type 2 Diabetes

It has been proposed that the increased presence of non-esterified fatty acids (NEFAs) is one of the major factors linking obesity and type 2 DM [21,22]. Adipose tissues, which are abundant in obese patients, exert various effects on the metabolic system by producing substances including cytokines, adiponectin, leptin, proinflammatory substances and NEFAs [23]. The continuous exposure to NEFAs, which natively act as a major insulin release stimulator, leads to pancreatic beta cell fatigue by facilitating significant malfunctions in glucose-stimulated insulin secretion pathways and the reduction of insulin biosynthesis [24]. The molecular mechanism is understood through increased activation of apoptotic unfolded protein response (UPR) pathways within the endoplasmic reticulum of the beta cells caused by increased presence of free fatty acids [24]. This leads to amplified biosynthesis of proinsulin and islet amyloid polypeptides (IAAP), followed by increased production of misfolded insulin and ROS. The grand effect is altered physiological endoplasmic reticulum calcium mobilization with increased proapoptotic signaling levels, proinsulin messenger ribonucleic acid (mRNA) degradation, and enhancement of local islet inflammation, culminating in malfunctions of pancreatic cells [25].

Toll-Like Receptors’ (TLR) role in insulin resistance formation poses another potential pathway linking obesity and type 2 DM. Recent studies have shown that the increased expression of TLR4 in adipocytes negatively affects insulin sensitivity through activation of metabolic endotoxemia in metabolic tissues [26,27]. Holland et al. found that TLR4 is a necessary signaling component for saturated-fatty-acid-induced ceramide biosynthesis, mediated by the upregulation of ceramide biosynthesis genes dependent on proinflammatory kinase IKKβ [28]. This ceramide biosynthesis is essential for TLR4-dependent insulin resistance [28]. Hussey et al. found that increased circulating levels of NEFAs resulted in increased expression of TLR3, TLR4, TLR5, several mitogen-activated kinases (MAPK), and inhibitor of kappa B (IκB) kinase (IKK)-nuclear factor kB (NFκB) [29]. IKKβ phosphorylates IκB, which releases NFκB from sequestration to allow for NFκB to translocate into the nucleus and activate proinflammatory genes [28]. Furthermore, Hussey et al. suggest that NFκB and MAPK may play a role in insulin resistance in muscle [29]. This research reinforces the hypothesis that NEFAs function as TLR4 ligands, stimulating NFκB and MAPK pathways to impair insulin signaling [29]. This mechanism also reinforces the chronic inflammatory state induced by obesity and type 2 DM.

Wang et al. found that TLR4/NFκB promote breast cancer progression through a resistin-mediated pathway [30]. TLR4 functions as a receptor for resistin, an adipokine related to obesity and T2DM which has been implicated in tumorigenesis through PI3K-AKT and MAPK pathways and is suspected to confer resistance to chemotherapy in breast cancer cells [31,32,33]. Like resistin, more research is needed to evaluate the role of NEFAs as ligands in the TLR4/NFκB signaling pathway for promoting breast cancer progression.

## 3. Obesity and Oxidative Stress

Obesity correlates with overactivation of proinflammatory cytokines and immunologic cells, such as macrophages, neutrophils, and B and T cells present within adipose tissues. The number of macrophages in obese individuals is increased by as much as three times that of a lean individual [34,35,36]. The proposed mechanism behind the impaired response of the immune system of an obese individual is the overloading of immunologic response caused by chronic inflammatory state. In adipose tissue, M2-type macrophages and other players with anti-inflammatory roles such as T helper 2 cells (Th2), regulatory T cells (Treg), and type 2 innate lymphoid cells (IlC2) are replaced by an increased number of proinflammatory M1-type macrophages, T helper 1 cells (Th1) and T helper 17 cells (Th17) [35,37]. The activation of immunologic cells within adipose tissue increases proinflammatory cytokines IL-6, IL-1β, and TNF-α [38]. This imbalance triggers regulatory pathways and limits the response of other acute triggers as well as potentiates the chance of acquiring type 2 DM via increasing NEFA levels [39,40,41]. As discussed previously, the increased presence of proinflammatory substances and immunologic cells due to an increased number of adipose tissues can eventually lead to type 2 DM due to various pathways surrounding pancreatic beta cells and their fragility.

Furthermore, studies have shown the significant impairment of antigen-specific T-cell responses in obese *murine* models, with increased susceptibility to *Listeria monocytogenes*, *Mycobacterium tuberculosis* (*M. tb*), *Klebsiella*, and *Streptococcus* [42,43]. Further data collection also indicates the linkage between greater morbidity caused by H1N1 influenza virus in obese patients compared to that of the normal population due to the defects in dendritic cells, leading to the ineffectiveness in antigen presentation to T cells and the subsequent inability to generate functional memory T-cell responses [44,45].

Oxidative stress plays a vital role in the development and progression of obesity. Studies have found that oxidative stress links obesity to several of its complications including respiratory issues, sleep apnea, and infertility [46]. Oxidative stress is both a stimulus and consequence of obesity. A diet high in dietary saturated fatty acids, trans fatty acids, and carbohydrates triggers biochemical pathways which increase ROS [47]. One such pathway generates ROS from Nicotinamide adenine dinucleotide phosphate oxidases (Nox). This coupled with endoplasmic reticulum stress in adipocytes leads to ROS [48]. ROS can in turn also lead to obesity through its stimulation of white adipose tissue deposition. Several studies have shown that oxidative stress leads to preadipocyte proliferation as well as an increase in the size of mature adipocytes [13].

ROS overproduction in obesity contributes to the hyperglycemia seen in this population. Several biochemical pathways upregulated by oxidative stress can inhibit Insulin Receptor 1 (IR1) through phosphorylation. One pathway involves c-jun N-terminal kinase 1 (JNK1), which disrupts downstream insulin signaling by phosphorylating IR1 at its inhibitory sites. This plays a key role in the development of insulin resistance. Oxidative stress may play a more direct role in insulin resistance by creating a state of chronic inflammation [49]. This upregulates the proinflammatory cytokines tumor necrosis factor alpha (TNFα) and interleukin-6 (IL-6) which phosphorylate IR1, also inhibiting insulin signaling [50].

Inflammation upregulates another important pathway leading to insulin resistance and hyperglycemia. The key player in the pathway involves inducible nitric oxide synthase (iNOS). A state of inflammation increases iNOS synthesis of nitric oxide up to a thousandfold [51]. When these high levels of nitric oxide combine with reactive oxygen species, they form highly reactive nitric-oxide-related species. Nitric-oxide-related species induce posttranslational modification of proteins including thiol nitrosylation (S-nitrosylation of cysteine residues) and tyrosine nitration [52]. A study on diabetic mice revealed increased expression of S-nitrosylated proteins, which was shown to contribute to insulin resistance [53].

Environmentally, evidence shows that increased consumption of foods rich in advanced glycation end products (AGEs) increase obesity, oxidative stress, and inflammation. AGEs are formed by reactive sugars and proteins in foods that are fried, roasted, grilled, or baked. AGEs mediate oxidative stress by binding to the AGE receptor (RAGE), which induces a signaling cascade involving p38 and JNK MAP, janus kinase–signal transducer and activator of transcription (JAK-STAT), and Cdc42/Rac pathways, which result in increased oxidative stress [54]. RAGE signaling also activates NFκB, furthering development of inflammation [54]. There is also evidence that decreased intake of AGEs reduces oxidative stress in people with diabetes [54]. Monden et al. revealed that increased RAGE expression is associated with adipocyte hypertrophy, suppression of glucose transporter type 4 (GLUT-4) and adiponectin, insulin resistance, and reduced insulin receptor substrate-1 (IRS-1) phosphorylation [55].

Oxidative stress protective mechanisms are also reduced in obesity. Bougoulia et al. found that glutathione peroxidase (GPX), a ubiquitously found antioxidant enzyme, is significantly reduced in obese women when compared to controls [56]. They also found that weight loss led to significantly increased levels of GPX [56].

Crown-like structures of the breast (CLS-B), formed when adipocyte hypertrophy causes necrosis enclosure by macrophages in a crown-like pattern, have emerged as histologic markers for local adipose tissue inflammation with a potential role in breast cancer development and prognosis. [57]. Studies have demonstrated that CLS-B creates a microenvironment in breast adipose tissue rich in proinflammatory cytokines and increases aromatization. More specifically, CLS is associated with TNF-α, inducible nitric oxide synthase (iNOS), CCL2, and IFN-γ [58]. This is believed to occur through macrophage exposure to saturated fatty acids from lipolysis which subsequently activate TLR4 at the macrophage cell surface, resulting in NF-kB activation and proinflammatory cytokine production [59,60,61]. CLS-B has also been associated with elevated levels of glucose, insulin, leptin, triglycerides, high-sensitivity C-reactive protein (CRP), and the proinflammatory cytokine IL-6 [62]. More research is required to explore the role of CLS-B in breast cancer pathogenesis as current studies linking CLS-B to breast cancer are limited.

## 4. Association between BMI and Breast Cancer

As the prevalence of obesity in the United States increases, there has been recognition of obesity as a risk factor for several cancers, including breast cancer. Most studies investigating this relationship have used BMI, calculated as weight divided by height in meters squared (kg/m^2^), as a measurement for obesity. BMI categories, as defined by the World Health Organization, are as follows: Overweight (25.0–29.9 kg/m^2^), Obesity class I (30.0–34.9 kg/m^2^), Obesity class II (35.0–39.9 kg/m^2^), and Obesity class III (≥40 kg/m^2^).

Although BMI remains widely used, there is concern about the utility of BMI in accurately assessing health. BMI measurements do not discriminate between muscle and adiposity, making its use unreliable in certain populations. Furthermore, the current WHO guidelines for BMI are largely based on white populations without consideration for varying muscle mass or body fat percentage variation between different racial and ethnic groups. Of importance is a study by Iyengar et al. that specifically investigates the utility of BMI as a measurement for risk assessment of breast cancer [63]. Using the WHI clinical trial data, this study examined postmenopausal women with normal BMI (18.5–24.9). It found that despite normal BMI, high body fat levels were associated with an elevated risk of invasive breast cancer [63]. Women with normal BMIs were found to have a 56% increase in breast cancer risk for every 5 kg increase in trunk fat. Specifically, these women were found to have an increase in ER-positive breast cancer. From this study, the authors conclude that, despite the utility of BMI in assessing risk of breast cancer in overweight and obese women, BMI categorization may be a poor metric for assessing risk in postmenopausal women who have a normal BMI. In The Cancer Prevention Study II (CPS II), waist circumference was also estimated as a risk for postmenopausal breast cancer. Although it did correlate with increased risk, it was not improved over BMI. Interestingly, a meta-analysis of studies reported for adjusting BMI found that waist circumference may be specifically associated with breast cancer risk in premenopausal women [64]. This association may also contribute to the possibility that elevated BMI in premenopausal women correlates with a lower risk of developing breast cancer, not due to adiposity, but rather due to increased muscle mass.

It has been established that increased BMI, whether classified as overweight or obese, is a modifiable risk factor for the development of breast cancer in postmenopausal women. Specifically, it has been reported by the American Cancer Society that 26,780 cases of breast cancer annually are attributable to excess body weight [65]. A study by Neuhouser et al. used data from the Women’s Health Initiative (WHI) clinical trials, which included 67,142 postmenopausal women enrolled over a period of five years [66]. The results from this study demonstrated that overweight and obese women had an increased risk of invasive breast cancer compared to their normal-weight counterparts. This study also demonstrated the relationship of increasing BMI with breast cancer, as women with BMI > 35 kg/m^2^ had the greatest risk of breast cancer development, with features of advanced disease and increased tumor size [66]. The results from the WHI clinical trials differ slightly from those of another large study in the National Surgical Adjuvant Breast and Bowel Project Breast Cancer Prevention Trial (NSABP P-1) and the Study of Tamoxifen and Raloxifene (STAR). These studies only found a nonsignificant increase in breast cancer in postmenopausal women with BMI > 30 kg/m^2^ compared to women with BMI < 25 kg/m^2^ [66,67]. Other studies that had similar findings of increased risk of breast cancer with increased BMI found that this risk typically manifests in “younger” postmenopausal women (mid–late 50s) [14,68]. NSABP P-1 and STAR found that elevated HRs were seen in premenopausal women for both ER-positive and ER-negative breast cancer; however, only ER-positive showed a statistically significant trend. Among postmenopausal women, there was a nonsignificant positive association between BMI and ER-positive breast cancer and no association between BMI and ER-negative breast cancer [67]. It is worth mentioning that these results differ from other studies which demonstrate a direct association between abdominal adiposity and ER-negative breast cancer only [69].

The influence of BMI on breast cancer risk is further increased in women who had a greater familial risk of breast cancer [14]. Increased breast cancer mortality with obese BMI has also been reported [66,70]. Despite the established relationship between increased BMI and breast cancer risk, the exact risk associated with increasing BMI has not been clearly quantified. In the study by Neuhouser et al., a strong linear trend is described where progressive increases in BMI correlate with increased risk, with a reported 58% increased risk in women with BMI > 35 kg/m^2^ [66]. Another study has attempted to quantify this relationship and suggests that there is a 2% increase in breast cancer for every 5 kg/m^2^ increase in BMI [71].

The predominant theory for the connection between obesity and breast cancer in postmenopausal women is the aromatization of adrenal androgens to estrogen adipocytes [66,67,72]. Consistent with this theory is the frequently reported association of estrogen receptor (ER)-positive breast cancers in overweight/obese women and lack of association with ER-negative breast cancers [66,67]. Although not explicitly studied in relation to breast cancer, leptin, which can increase estrogen levels, is increased in overweight and obese individuals [66]. Further supporting the hormone theory is the attenuation of the association with BMI and breast cancer risk in women with a history of postmenopausal therapy treatment [67]. Other theories of the relationship between breast cancer and BMI include insulin resistance in this population and insulin-like growth factor-1 (IGF-1) upregulation leading to cell proliferation and increased free estrogen levels through a reduction in sex-hormone-binding globulin [72]. Decreased adiponectin in obese individuals has also been suggested as a cause for high estrogen levels [72].

In contrast to postmenopausal women, the relationship between increased BMI and breast cancer risk appears to be inverse in premenopausal women [14,15,16]. Schoemaker et al. found a linear association between decreased breast cancer risk and increased BMI in premenopausal women with an estimated risk reduction of 23% for every five-unit difference for BMI at ages 18–24 years old and 12% for BMI at 45–54 years old [16]. Other studies have reported similar reductions in risk but to a lesser degree of 5–9% in risk reduction [73]. Additionally, the study by Hopper et al. found that greater BMI at young age is associated with a decreased risk of breast cancer, although this negative association does not have a substantial influence on absolute risk of breast cancer [14]. Theories behind this association are not well established. There is reason to think it may be estrogen-related, as those with higher BMIs have longer and more irregular menstrual cycles with more frequent anovulation and hence decreased estrogen and progesterone levels. This theory, however, is contraindicated by the protective effects of BMI persisting in women with no history of infertility due to ovulatory disorders [15]. Another theory proposes that increased estrogen, because of childhood adiposity, leads to increased expression of tumor-suppressing genes and earlier breast differentiation [16].

## 5. Pathophysiology of Breast Cancer

Breast cancer is characterized by uncontrolled proliferation and growth of abnormal cells in the breast. Most breast cancers are hormone-dependent, and as such, classification is commonly based on expression of the following hormone receptors: estrogen (ER), progesterone (PR), human epidermal growth factors (HER2), or none, termed “triple-negative breast cancer” [74]. Its pathophysiology is complex and multifactorial, arising from genetic, hormonal, and environmental factors. Of these, hormonal and environmental factors are often tied to obesity [75].

At the genetic and molecular levels, ROS from oxidative stress initiates the process of cancer by damaging cellular components, such as deoxyribonucleic acid (DNA), proteins, and lipids, ultimately inducing genetic mutations. This triggers the activation of signaling pathways that promote cancer cell growth and survival, stimulating angiogenesis and metastasis, and impairing the immune system’s ability to recognize and eliminate cancer cells [76,77]. Several studies show that breast cancer cells generate higher levels of ROS and decreased antioxidants compared to normal breast cells, suggesting that they are under increased oxidative stress. This increased ROS production has been linked to the activation of several oncogenes and the inactivation of tumor-suppressing genes, which are key factors in the development and progression of breast cancer [78]. For example, ROS aids in the survival of cancer cells by activating NF-kB, matrix metalloproteinases (MMPs), and VEGF-dependent angiogenesis and metastasis via MAPK/ERK1/2, p38, JNK, and PI3K/Akt pathways when exposed at a moderate concentration. ROS also increases invasion and angiogenesis by increasing expression of hypoxia-inducible factor-1 (HIF-1a), further triggering NF-kB and upregulating the transcription of programmed cell death ligand 1 (PD-L1) and release of chemokines [79]. ROS have also been shown to increase expression of the aryl hydrocarbon receptor (AhR), which enhances tumor growth and proangiogenic function of macrophages through further chemokine production. High ROS concentrations, however, can cause cancer cell apoptosis, demonstrating the importance of the tumor ROS microenvironment in ROS regulation [79,80]. In addition, TNBC shows increased expression of glutamine transporter SNAT2/SLC38A1, providing oxidative stress resistance [81].

Multiple studies further prove that a reduction in oxidative stress improves breast cancer prognosis, though the type of antioxidant is relevant [82]. Nanoencapsulated antioxidants, such as resveratrol, a natural polyphenol mostly found in the skin of grapes [83], and chia oil, an omega-3 fatty acid, have been shown to reduce progression of breast cancer via regulation of oxidative stress [84,85,86], with the former being tested in humans and the latter being tested in *murine* models. In the context of oxidative stress as a modifiable environmental factor, a randomized clinical trial has shown that long-term yoga significantly reduces inflammatory cytokines and oxidative stress specifically in breast cancer patients [87].

At the metabolic level, epidemiological studies, mostly done on type 2 DM patients due to relatively higher disease prevalence, show that diabetic patients have worse outcomes across multiple tumor types [88,89]. Of note, diabetic conditions that may confound each other include hyperglycemia, hyperinsulinemia, and chronic inflammation. However, several studies have shown hyperglycemia as an independent risk factor for multiple cancers [90,91]. In general, a high rate of glucose uptake is central to the metabolism of neoplastic tumor proliferation [92]. In addition, transient exposure to hyperglycemic conditions showed permanent aggressive traits of tumor growth via epigenetic modifications, as well as increased tissue-specific angiogenesis [93,94,95]. Specific to breast cancer cells, high carbohydrate levels enhance expression of the neuregulin 1-gene, an endogenous ligand for the HER3 receptor [96], as well as multiple glucose-metabolism-related long-coding ribonucleic acids (lncRNAs) that increased risk of breast cancer progression [97]. Of note, expression of HER3 has been linked to resistance to drugs that target other HER receptors, such as agents that act on EGFR or HER2 [98].

Hyperinsulinemia, which results from elevated glucose levels, has also been shown to play a role in cancer development. Insulin receptors are elevated in breast cancer, indicating a potential sensitivity of tumor cells to a high insulin state [99]. Insulin-like growth factors 1 and 2 (IGF-1 and IGF-II) are peptides with similar actions to insulin but are secreted in response to growth hormones. These hormones play a role in lowering blood glucose levels. IGF-II levels were studied in obese individuals before and after gastric banding, which reduced their BMI. It was found that IGF-II levels were significantly higher when the subjects were morbidly obese [100]. The tyrosine kinase receptors of this hormone play an important role in promoting cellular proliferation such as in fetal development. The receptors in cancerous breast tissue have been found to be sixfold higher than in normal breast tissue [99].

Another hormonal factor that contributes to the development of breast cancer is estrogen, which plays a crucial role in the progression of breast cancer, as most tumors are initially hormone-dependent [101,102]. However, breast cancer cells may become hormone-independent, as mutations in the estrogen receptor itself cause a lack of response to estrogen [103]. Thus, adipose-derived estrogens, often elevated in obesity or high-body-fat women, may increase their risk for estrogen-receptor-dependent breast cancer [63]. This is particularly important in postmenopausal obese women, where estrogen synthesis after menopause takes place almost entirely in adipose tissue.

Although TNBC cells do not express ERs, they are estrogen-responsive via ER-independent pathways. It has been demonstrated that increasing levels of circulating estrogens was sufficient to promote the pathogenesis of ER-negative cancers, including TNBC [104]. Research has also shown that estrogen increases host angiogenesis by mobilizing and recruiting bone-marrow-stromal derived cells into sites of angiogenesis, as well as directly to the growing tumor mass. Therefore, estrogen may promote growth of ER-negative breast cancers such as TNBC by also acting on cells distinct from the cancer cells to stimulate angiogenesis [104]. ERβ, GPER, as well as estrogen-related receptors (ERRs) are frequently expressed in TNBC. ERβ is a tumor suppressor which has been shown to be of major importance for estrogen action in TNBC, with expression found in 30–60% of TNBC cases [105]. It has been shown that cell lines with mutant TP53 expressing high ERβ levels have better survival than TNBCs without TP53 mutation [105]. The clinical relevance of GPER in TNBC remains controversial, with some studies citing antitumor properties and others claiming tumor promotion. Recently, Pal et al. identified a significant positive association between GPER and ERα expression in breast tumors. Interestingly, they found that high expression of GPER is associated with significantly longer overall survival (OS) in ERα-positive patients. Conversely, they found poor OS in ERα-negative patients. These results require further investigation, however, as ERα+ patients generally have better OS. Importantly, since GPER is downstream of estrogen-ERα signaling, it may serve as a marker for distinguishing endocrine-responsive breast cancers. More research is necessary to evaluate the paradoxical role GPER may serve in breast cancer, as research shows it both promotes and inhibits cell proliferation [106]. In the absence of ERα, such as in TNBC, ERRα becomes a major regulator of genes containing estrogen response elements (EREs). While ERRs do not bind estrogens, they are able to regulate estrogen signaling by constitutive activity and bind compounds with estrogen-like structures [105]. ERRα expression in TNBC is associated with worse prognosis. However, an improved outcome was seen in TNBC patients with high ERRα expression who were treated with tamoxifen [107]. Together, these results demonstrate that despite the absence of ERα in TNBC, the activation of ERβ and targeting of ERRs and GPER may serve as important endocrine therapies for TNBC treatment [105].

Increased circulating ROS, proinflammatory cytokines, hyperglycemia, hyperinsulinemia, and increased estrogen synthesis resulting from obesity contribute to the pathophysiology of breast cancer development. The impact of cholesterol imbalance in obese individuals also lends to the pathogenesis of breast cancer. Research shows that both TNBC and HER2+ breast cancer cell lines display enhanced proliferation and invasion in the presence of circulating LDL-C [108,109,110]. Interestingly, HDL is also implicated as breast cancer cells express scavenger receptor BI proteins (SR-BI) in excess, increasing cholesterol absorption and subsequent tumorigenesis [111]. Research has also shown that the liver x receptor (LXR), a regulator of cholesterol metabolism, is increased in TNBC when compared to other subtypes. Lianto et al. identified LXR splice variants that shorten disease-free state (LXRα1 and 4) and variants that increase disease-free state (LXRβ and α with truncated ligand-binding domains) [112].

## 6. The Role of the mTOR Signaling Pathway in Breast Cancer

In obese individuals, high levels of NEFAs, glycerol, hormones, and proinflammatory cytokines are released from adipose tissue, leading to adipose dysregulation and oxidative stress [113]. The oxidative stress leads to posttranslational modification in signaling proteins and ultimately insulin resistance [114]. Insulin resistance can lead to impairments in glucose utilization and metabolic consequences of hyperglycemia [115]. This puts obese individuals at high risk for diabetes and many other comorbid chronic conditions including stroke, cardiovascular disease, poor mental health, and cancer [2]. The onset of obesity is linked to an increased occurrence of aggressive, metastatic phenotype of cancer [116]. Obesity is associated with the worst disease-free survival and overall survival of all breast cancer subtypes [117]. Breast tissue is rich in adipocytes which, in times of dysregulation seen in obesity, secrete many inflammatory mediators that induce tissue damage and potentiate cancer progression [116]. The dysregulation of adipocytes also leads to a hyperglycemic environment, which can also help cancer cells survive.

The preferential utilization of glucose for energy by cancer cells was first described by Otto Warburg in the early 1900s [118]. He observed cancer cells utilizing more glucose than the surrounding tissue and forming lactate even in the presence of oxygen and functioning mitochondria. This was in stark contrast to healthy cells, which primarily utilize oxidative phosphorylation for a more efficient adenosine triphosphate (ATP) generation and only undergo glycolysis in anaerobic conditions [119]. This seemingly inefficient nonhypoxic glycolysis has been termed the Warburg Effect. Recent genetic studies have shown that the Warburg Effect is critical for tumor cell proliferation [120]. Although more ATP is generated through oxidative phosphorylation, one study has proposed that cells with a higher rate, but lower yield, of ATP may have an advantage when competing for resources [121]. Tumor cells have many competitors in their microenvironments including stromal and immune cells, and their preferential affinity for glucose may help them in being the first to acquire the important resource [122]. Disproportionately higher levels of glucose uptake in cancer cells have permitted the detection and monitoring of cancer through [18F] fluoro-2-deoxyglucose (FDG) PET scans. Intracellular FDG levels have a direct correlation to glucose uptake in cancer cells [123]. Further illustrating the reliance of tumor cells on glucose is a study which showed that fasting, and thus glucose-restricted, mice had increased response to chemotherapy with increased cancer-free survival [124].

mTOR is a serine/threonine kinase which exists as mTOR complex 1 (mTORC1) and 2 (mTORC2). Its activation involves the phosphoinositide 3-kinase/protein kinase B (PI3K/AKT) pathway, which is activated by tyrosine kinase receptors, insulin receptor substrate 1 or 2 (IRS-1 or IRS-2), RAS protein, and several other G-protein-coupled receptors. mTORC2 plays an important role in full activation of AKT, thus inhibiting proteolysis of Cyclin D1/E, as well as the activation of serum- and glucocorticoid-inducible kinase 3 (SGK3) [125,126]. Cyclin D1 is implicated in tumorigenesis, and Cyclin E is a prognostic marker in breast cancer [127]. Following activation by the PI3K/AKT pathway, mTORC1 regulates the assembly of the eukaryotic translation initiation factor 4F (elF4F) complex by mediating phosphorylation of 4E-binding proteins (4EBP1), as well as the activation of ribosomal protein S6 kinase 1 (p70S6K) [128]. 4EBP1 dissociates from elF4E, which is then free to bind to elF4G, resulting in elF4F formation [128]. In addition to mTORC1, p38 MAPK signaling pathways activate MAPK-interacting kinases (MNKs), which bind to elF4G and phosphorylates elF4E [128]. The MYC oncoprotein mRNA is a translational target of the elF4F complex. MYC also promotes elF4E transcription, further upregulating its own translation [128]. Both elF4E and elF4F are implicated in neoplasia. It is hypothesized that elF4E mediates tumorigenesis in cancers with uncontrolled mTORC1 activation, as seen in patients with phosphatidylinositol 3-kinase (PI3K) activation mutations, phosphatase and tensin homolog (PTEN) loss, as well as deletions in tumor suppressor proteins tuberous sclerosis complex (TSC1 and TSC2) [125,129]. elF4E is overexpressed in a range of human cancers, including breast cancer, and in most cases is associated with increased recurrence and decreased survival [128]. Furthermore, Zindy et al. demonstrated that the elF4F complex may be a vital mediator of breast cancer resistance to anticancer drugs that target HER2 and EGFR [130].

Uncontrolled activation of the mTORC1 pathway has been well studied in many pathologies, including type 2 DM, cancer, and obesity. Upregulation of this pathway leads to induction of many metabolic processes, including the glucose utilization in cancer cells [131]. Specifically, the mTORC1–p70S5K pathway regulates protein and lipid synthesis, cell growth, size, and metabolism, and plays an important role in obesity, diabetes, and cancer [132]. Studies have shown that inhibition of the mTORC1 pathway suppresses the localization of glucose transporter-1 (GLUT1), a critical component of glycolysis, and that activation of mTORC1 stimulates glycolysis and glucose uptake via hypoxia-inducible factor (HIF1α) [131,133]. mTORC1 also plays an important role in adipogenesis and the maintenance of adipocytes [134]. Adipose tissue is the essential organ leading to obesity and insulin resistance [135]. The mTORC1–p70S6K pathway phosphorylates glutamylprolyl-tRNA synthetase (EPRS), which subsequently is released from amino acyl tRNA multisynthetase and interacts with fatty acid transport protein 1 (FATp1), promoting its translocation to the plasma membrane for fatty acid uptake [136]. mTORC1 also stimulates de novo lipogenesis through the mTORC1–p70S6K pathway and by modulating Lipin 1 localization and sterol regulatory element-binding protein 1 (SREBP1) expression, as seen in Figure 1 [137]. One study showed that adipocyte-specific mTOR knockout mice had decreased adipose tissue mass and insulin resistance, further elucidating a relationship between mTOR and obesity [138]. Xiang et al. found that mTORC1 induced phenotypic class switching of brown adipose tissue (BAT) to white adipose tissue (WAT), a process which is reversible utilizing mTORC1 inhibitor rapamycin [139]. mTORC1 and its co-protein, Raptor, are also implicated in phenotypic switching from WAT to BAT, mediated by β3 adrenergic signaling [140]. In addition to mTORC1’s role in lipogenesis and phenotypic switching, mTORC2 and its co-protein, Rictor, promote de novo lipogenesis and hepatic glucose metabolism by upregulating lipogenic transcription factor carbohydrate response element-binding protein (ChREBPβ) [141]. Obesity reciprocally induces a hyperactivation of the mTOR pathway [142]. This dysregulation may facilitate the development of type 2 DM and cancer in obese individuals. The cell signaling pathways induced by obesity create an environment where tumor cells can proliferate and thrive by increasing preferential usage of aerobic glycolysis.

## 7. The Role of Glutathione in Breast Cancer Therapy

Glutathione (GSH) is a tripeptide that plays key roles in regulating metabolic processes, especially tumorigenesis and cancer pathophysiology. Its structure of glutamic acid, cysteine, and glycine, respectively, plays a key role in DNA-binding activity in the thiolation of proteins [143]. For instance, the common tumor suppressor p53 has 12 cysteine residues in its structure, and oxidizing these residues will inhibit p53 function [144]. GSH acts as a regulator in cell differentiation, proliferation, and maintenance of intracellular redox homeostasis as well as protecting the cell from the overexpression of ROS. ROS are involved in intracellular signaling, but the overproduction of ROS (also called oxidative stress) can lead to cell dysfunction and death. This oxidative stress can be mitigated by GSH as an antioxidant agent.

Despite the antioxidant properties of GSH, elevated levels of GSH have been found in breast cancer. This phenomenon occurs due to several interacting factors. The higher levels of ROS found in tumor cells require a greater concentration of GSH to attempt to alleviate oxidative stress in growing tumor cells. This regulation activates GSH synthesis by the expression of cystine/glutamate antiporter (Xc-) [145]. In addition, estrogen-related receptors (ERRs) play a role in sensing and regulating ROS production in breast cancer and depend on GSH synthesis [146].

GSH depletion has been evaluated as a possible strategy to increase oxidative stress to eliminate tumor cells. This technique involves generating ROS in tumor tissues, decreasing GSH detoxification, and inducing iron-dependent cell death through ferroptosis [147]. Reducing ROS scavenging combined with GSH depletion has been found to have better therapeutic outcomes in terms of tumor weight and apoptosis levels in tumor tissues compared to ROS-based therapy alone [148]. As for reducing GSH detoxification, it has been found that adding a GSH synthesis inhibitor significantly increased cell apoptosis induced by cisplatin, thus increasing the efficacy of treatment in drug-resistant tumors [149]. Finally, ferroptosis involves the accumulation of Lipid hydroperoxides and can be initiated by GSH depletion via dysregulating the cystine/glutamate antiporter and glutathione peroxidase 4 (GPX4) [150]. Sulfasalazine and erastin are both Xc- inhibitors that have shown an effect on apoptosis-inducing ligands in colon cancer cells [151]. Erastin specifically kills tumor cells without damaging normal cells, which aids in decreasing tumor resistance [152].

For patients with triple-negative breast cancer undergoing chemotherapy, there are elevated levels of GSH along with increased expression of GCLM and SLC7A11 (the protein for cystine/glutamate antiporter). By inhibiting GSH synthesis, it would inactivate breast cancer stem cell phenotypes and subsequent copper chelation. This inhibition of copper chelation induces downstream MAPK activity and forkhead box class O 3 (FoxO3) dephosphorylation, which would stunt breast cancer cell growth and spread [153]. By depleting GSH in breast cancer therapy, there is a greater ability to cause ferroptosis and thereby improve chemotherapy outcomes.

The most common GSH synthesis inhibitor is buthionine sulfoximine (BSO), which irreversibly inhibits the first step in GSH synthesis. BSO has been shown to increase the efficacy of cisplatin and increase ROS accumulation for further tumor cell death [154,155]. The latter process occurs through BSO-activated GSH depletion that further induces ferroptosis [156]. However, the biggest obstacle that is faced by GSH depletion treatment in breast cancer patients is the short half-life of BSO, which requires continued infusions and a risk of thrombocytopenia and leukopenia [157].

## 8. Obesity Targeting Therapies for Breast Cancer

For the last few decades, tamoxifen has played a significant part in the treatment of ER-positive breast cancer. However, tamoxifen resistance commonly occurs [158]. It has been shown that inhibition of mTOR activity restores the ability of Akt-overexpressing breast cancer cells to respond to tamoxifen [159]. In addition, mTOR is important due to its relationship with autophagy. In normal cells, autophagy typically provides protection against malignant transformation. However, in cancer cells, autophagy may allow tumor growth. It has been discussed that the transformation from a healthy cell to a cancer cell involves a temporary loss of autophagic competence [160]. Initially, defects in autophagy allow a healthy cell to acquire malignant features. Eventually, when autophagy is restored, the cancer cell utilizes this to support its survival and growth. The mTOR complex 1 (mTORC1) is the major regulator of autophagy in breast cancer cells [161,162,163].

Rapamycin and its analogs, including everolimus and temsirolimus, allosterically inhibit mTORC1. Their efficacy is also limited by resistance. One prevailing hypothesis centered around rapamycin resistance is activation of AKT through loss of mTORC1/p70S6K1-dependent negative feedback loop resulting in only weak inhibition of 4EBP1 protein phosphorylation. Evidence exists that AKT mediates the weak inhibition of 4EBP1, although the mechanism is unclear [125]. Wang et al. identified glucocorticoid-regulated kinase 3 (SGK3) as a mediator of 4EBP1 reactivation [125]. In addition, they found that SGK3 deletion in combination with AKT inhibition significantly improved rapamycin efficacy by reducing 4EBP1 reactivation in a breast cancer *murine* model [125]. Identifying SGK3 as a novel mediator of rapamycin resistance reveals novel adjunctive therapies to improve efficacy in patients with uncontrolled mTORC1 activation, as seen in Figure 2.

The BOLERO-2 trial by Baselga et al. analyzed the effects of everolimus and exemestane versus exemestane and placebo for patients with hormone-receptor-positive advanced breast cancer. Everolimus selectively inhibits mTORC1, and exemestane inhibits aromatase. Everolimus plus exemestane produced a median progression-free survival of 6.9 months, whereas placebo plus exemestane produced a median progression-free survival of 2.8 months [164]. This study demonstrates that the use of an mTOR inhibitor significantly improves clinical outcomes, further highlighting how this route of pharmacotherapy may be beneficial to treating future resistant breast cancers.

The PrE0102 phase II trial by Kornblum et al. compared the effects of fulvestrant plus everolimus versus fulvestrant plus placebo in women with hormone-receptor-positive metastatic breast cancer resistant to aromatase inhibitor therapy. Fulvestrant blocks estrogen binding and downregulates estrogen receptors. The addition of everolimus to fulvestrant improved the median progression-free survival from 5.1 months to 10.3 months [165]. This study further supports the idea that the use of an mTOR significantly improves clinical outcomes.

The MANTA phase II trial by Schmid et al. examined the effects of fulvestrant plus vistusertib versus fulvestrant plus everolimus versus fulvestrant alone for women with hormone-receptor-positive metastatic breast cancer. Everolimus selectively inhibits mTORC1, whereas vistusertib inhibits both mTORC1 and mTORC2. Ultimately, groups treated with fulvestrant/everolimus demonstrated significantly longer progression-free survival (12.3 months) versus those treated with fulvestrant/intermittent vistusertib (8.0 months), with fulvestrant/daily vistusertib (7.6 months), and with fulvestrant alone (5.4 months) [166]. This trial shows that mTORC1 inhibition is more critical in increasing survival, rather than both mTORC1 and mTORC2 inhibition. These studies indicate that everolimus demonstrates the greatest clinical benefit in the second-line setting.

Novel mTOR inhibitors, such as IBL-302, DHW-208, XS-2, and gedatolisib, are currently undergoing early-phase clinical trials. IBL-302 is a molecule designed to inhibit PIM kinase, pan-PI3K, and mTOR altogether [167]. DHW-208 is a 4-aminoquinazoline derivative that has shown dual inhibition of PI3K and mTOR in vitro [168]. XS-2 is a thiophene–triazine derivative that has also shown dual inhibition of PI3K and mTOR, in addition to low toxicity in in vivo experiments [169]. Gedatolisib is a potent, reversible dual inhibitor of PI3K and mTOR that was granted a Fast Track designation by the United States Food and Drug Administration in 2022. Currently, there are investigations on the combinations of gedatolisib and cofetuzumab pelidotin as well as gedatolisib plus fulvestrant, with or without palbociclib (VIKTORIA-1 phase III trial). Recent developments in breast cancer research indicate that mTOR inhibitors are a viable method of treatment.

Another target of focus for novel therapeutics is leptin, specifically the leptin receptor. Linares et al. found that leptin both increases proliferation of breast cancer cells in vitro and diminished the efficacy of tamoxifen [170]. Otvos Jr. et al. found that the leptin receptor antagonist peptide Allo-aca extended the average survival time from 15.4 days to 24 and 28.1 days at 0.1 and 1 mg/kg/day doses, respectively, in a triple-negative breast cancer mouse xenograft model. Comparatively, the conventional treatment of cisplatin at doses of 1 mg/kg/day prolonged the average survival time to 18.6 days. In normal mice, Allo-aca produced no systemic toxicity up to bolus doses of 50 mg/kg [171]. However, Allo-aca was found to cross the blood–brain barrier and produce orexigenic effects, leading to weight gain. As a result, Beccari et al. generated D-Ser, a leptin-based peptidomimetic with peripheral leptin receptor antagonistic activity [172].

Synthesis of a four-amino-acid peptide, Amino acids 39–42 (Leu-Asp-Phe-Ile) (LDFI), based on the wild-type sequence of leptin-binding site I, has been shown to inhibit breast cancer growth in vitro and in vivo [173]. In both ERα-positive and ERα-negative human breast cancer cells, LDFI inhibited leptin-induced proliferation, motility, and signaling activation. In xenograft models, LDFI significantly reduced breast tumor growth. The peptide produced no signs of systemic toxicity or significant effects on body weight in mice. The disadvantages of both Allo-aca and LDFI are the general short half-life of peptides in the human body. Overall, the focus on the leptin receptor for the treatment of breast cancer seems to be another viable method of pharmacotherapy.

Adiponectin is an adipokine with strong anti-inflammatory and proinsulin signaling effects and has been explored as a therapeutic target. In obese individuals, levels of adiponectin are reduced, placing individuals at risk of breast and other cancers [174]. Pham et al. found that supplementing globular adiponectin (gAcrp) decreases cellular lipid pools, induces lipolysis, triggers lipid raft disruption and apoptosis, and suppresses fatty acid synthesis via deacetylation of SREBP-1 in breast cancer cells. They further identified that SIRT-1, a master cellular metabolic regulator and downstream effector of adiponectin induction, is essential for lipid metabolic reprogramming and apoptosis induction by adiponectin. gAcrp also significantly suppressed phosphorylation of mTOR through SIRT-1 signaling [175]. In addition to adiponectin, PAI-1 is a serine protease inhibitor produced by adipocytes which also affects adipocyte differentiation and insulin signaling [176]. The overexpression of PAI-1 has been associated with numerous obesity-related cancers, including breast cancer [177]. More research is necessary to investigate the role of PAI-1 expression in breast cancer development in obesity and as a novel therapeutic target. PPARγ synthetic ligands such as rosiglitazone and pioglitazone have also been shown to increase adiponectin levels. More research is needed to explore the role of thiazolidinediones as adjuvants for treating obesity-driven cancers, as low adiponectin is associated with cancer progression [178].

Aromatase inhibitors (AIs) serve as an adjuvant endocrine therapy in postmenopausal HR-positive breast cancer patients. Pfeiler et al. found that AIs are less efficient at suppressing estradiol serum levels in obese women when compared to nonobese women. In addition to increased aromatization seen in obese patients, insulin resistance may counteract AI treatment through direct activation of the estrogen receptor by insulin [174,179]. AIs have been found to only be effective in postmenopausal women and provide only time-limited effects, as estrogen levels eventually return to normal once withdrawn [179].

Insulin receptor signaling pathways have also been explored for breast cancer treatment. Several IGF1R antibodies have been utilized with mixed results. Specifically, ganitumab in combination with the AI exemestane or the ER downregulator fulvestrant failed to improve progression-free survival (PFS) and induced significant rates of grade 3/4 hyperglycemia [180]. Cixutumumab, robatumumab, and istiratumab have also been investigated in Phase 1 and 2 clinical trials but have shown limited efficacy and poor tolerability [178]. Dusigitumab and Xentuzumab, monoclonal antibodies that bind IGF-2 and IGF-1/2, respectively, were also explored in clinical trials with no significant improvement. Metformin successfully attenuates expression of IGF1 and the activation of mTOR/Akt in breast tissue; however, results have been mixed and have not improved PFS in a metastatic setting. Still, metformin is being actively researched as an adjuvant in several clinical trials [178].

Lifestyle changes represent a targeted therapy for improving outcomes in breast cancer treatment. Studies show that a low-fat diet reduces the risk of invasive breast cancer. Restricted alcohol intake was also shown to be beneficial in several studies, perhaps due to excessive alcohol intake altering hormonal hemostasis [181].

## 9. Breast Cancer and Immunotherapy

As mentioned earlier, PD-L1 expression is increased by obesity, as well as its receptor, PD-1, which serve as immunosuppressive effects in tumors by inducing T-cell anergy. Novel immunotherapeutic methods involving monoclonal antibody immune checkpoint blockade (ICB) immunotherapy have successfully been shown to block PD-1 and PD-L1 to restore T-cell function and, as a result, weaken the tumor defense. Trials have shown that obese patients respond better to immunotherapy in melanoma, non-small-cell lung cancer, and renal cell carcinoma [182]. Pingili et al. explored the use of anti-PD-1 immunotherapy in obesity-associated breast cancer in murine models. They found that anti-PD-1: (1) decreased tumor burden, (2) reversed immunosuppression by increasing tumor dendritic cell content and M1 tumor-associated macrophages (TAMs) and decreasing immunosuppressive protumoral M2 macrophages, (3) increased antitumor responsiveness of M1 macrophages, dendritic cells, and CD8+ T cells, (4) and increased the diversity of the gut microbiome previously shown to influence ICB response in cancers [183]. In addition to these findings, they found that obesity increased the concentration of protumor immunosuppressive splenic myeloid-derived suppressor cells (MDSCs) and M2 macrophages, as well as bone marrow MDSCs, all of which were significantly reduced with anti-PD-1 therapy [183]. Although effective in reducing tumor burden in murine models, most breast cancer patients fail to respond to ICB, and more research is needed to identify additional immunotherapy targets or improve the adjuvant response in patients [184,185].

Khojandi et al. found that immunotherapy survival was predicted by oxidized LDL (ox-LDL) rather than BMI or serum leptin concentrations. Using *murine* models, they found that heme oxygenase-1 (HO-1), activated by ox-LDL, serves as a resistance mechanism to anti-PD-1 immunotherapy. Of note, ox-LDL also has an inverse correlation with CD8+ T-cell function. They found that female mice bearing breast tumors treated with anti-PD1 or an HO-1 inhibitor (iHO-1) experienced no benefit; however, mice treated with both experienced significantly reduced tumors by day 16 [186]. These data represent an important new target for improving patient response to ICB and require further research.

Novel research is identifying the gut microbiome as a model to predict patient responsiveness to immunotherapy. Studies have identified an improved response to anti-PD-L1 therapy in murine models harboring *Akkermansia muciniphila*, *Bifidobacterium longem*, *Collinsella aerofaciens*, and *Faecalibacterium prausnitzii* [187,188,189]. Other studies have shown that certain bacteria may be associated with immunotherapy toxicity, such as *Bacteroidaceae, Barnesiellaceae*, and *Rikenellaceae* [189]. As a result, gut microbiome may be utilized as an estimate of immunotherapy efficacy, and previous or recent antimicrobial usage may be of significant influence. *Bacillus fragilis* has been specifically implicated in breast cancer as a procarcinogenic gut microbe which is also strongly linked with obesity gut dysbiosis. Parida et al. demonstrated that enterotoxigenic *B. fragilis* (ETBF), which produces *B. fragilis* toxin (BFT), induces systemic inflammation, augments tumor immune microenvironment, and serves a permissive role in metastatic seeding to liver and lung tissues [190]. Targeting the gut microbiome may serve as a therapeutic adjunct in breast cancer treatment and prevention.

In obesity, natural killer (NK) and natural killer T (NKT) cells are significantly decreased in number and activity. Lynch et al. found that transferring invariant NK T (iNKT) cells in obese mice successfully reduced body weight, decreased triglyceride and leptin levels, as well as increased production of anti-inflammatory cytokines to dampen the metainflammatory state of the tumor microenvironment [58,191]. As obesity serves as a multifactorial influence on immune cell response, further research is needed to explore immunotherapy adjunct treatments alongside ICB to improve patient responses. Serum small nucleolar RNAs (snoRNAs) are associated with obesity and immune suppression. Specifically, SNORD46 has been identified in adipocytes and inhibits signaling pathways of IL-15, a growth factor for T and NK cells [192]. SNORD46 also inhibits lipolysis and adipose browning [192]. Zhang et al. investigated the interconnection between SNORD46, obesity, and breast cancer. They found that SNORD46 was significantly elevated in the serum of patients with BMI > 30. Additionally, adipose tissue expressed the highest density of SNORD46. SNORD46 was associated with less tumor infiltration of NK cells in breast cancer and fewer CD8+ T cells and NK cells in BRCA patients with elevated serum SNORD46 levels. They also found that targeting SNORD46 successfully restored antitumor immune response in NK cells, reducing tumor burden in colorectal adenocarcinoma. Interestingly, SNORD46 mutant mice led to improved glucose tolerance, decreased body weight, reduced liver steatosis and fatty acid accumulation, and decreased lipid accumulation in the liver, demonstrating that modifying SNORD46 may yield anticancer properties and anti-obesity properties [192].

## 10. Conclusions

There is sufficient evidence to show that obesity is a significant modifiable risk factor in the pathogenesis of type 2 DM and breast cancer. Increased oxidative stress, insulin resistance, hyperglycemia, and hyperlipidemia predispose the body to a pro-oncogenic state. Increased oxidative stress produced in states of obesity creates a feedforward cascade that furthers the development of obesity and type 2 DM. While metrics for the assessment of obesity such as BMI are inaccurate, there remains a trend in the risk of developing breast cancer which is reversed after menopause. Obesity in postmenopausal women is associated with an increased risk of developing breast cancer, while obesity in premenopausal women may confer a protective effect. This change may be due to variability in estrogen levels, which are implicated in estrogen-sensitive breast cancers and increased in states of heightened adiposity. Hyperglycemia maintains a favorable state for tumorigenesis, as cancer cells preferentially utilize glucose for aerobic glycolysis, effectively outcompeting noncancerous neighboring cells. The biochemical mechanisms that outline the development of cancer in states of metabolic imbalance are growing, with the mTOR signaling pathway and hormones such as leptin serving as scaffolds for pharmacological intervention. New research is identifying mechanisms by which the mTOR signaling pathway can resist inhibition and how novel mechanisms may be able to rescue efficacy, such as inhibiting AKT and SGK3, as well as mechanisms to antagonize overstimulation of the leptin receptor. Lastly, while GSH is often seen in literature serving as a protective role in inflammatory conditions, the inhibition of its synthesis may prove to be an effective adjunct to anticancer drugs by increasing damaging ROS within breast cancer cells.

## Figures and Tables

**Figure 1 cells-12-02061-f001:**
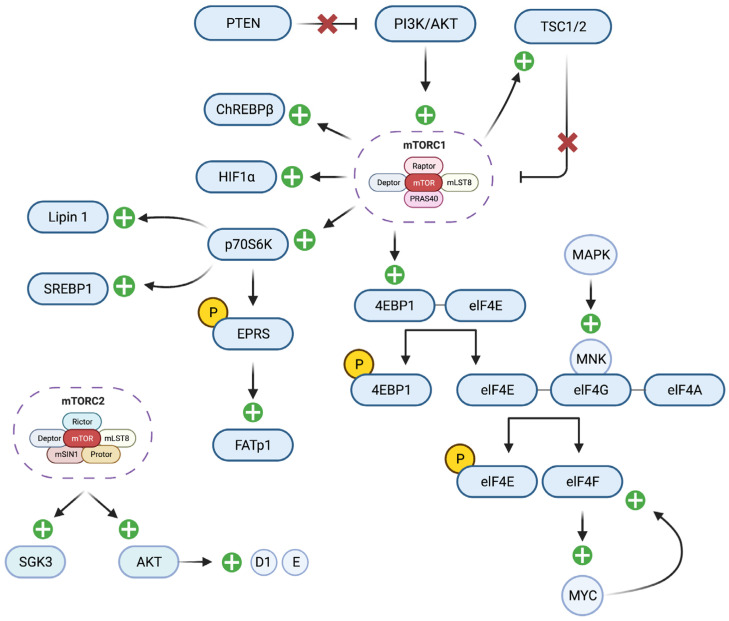
mTORC1 and mTORC2 in Tumorigenesis. mTORC1 phosphorylates 4EBP1, liberating elF4E to bind to elF4G and elF4A. MAPK activates MNK, which binds to elF4G and phosphorylates elF4E. elF4E is removed upon phosphorylation, leaving elF4G and elF4A as the elF4F complex. The elF4F complex increases translation of MYC, which triggers a feedforward loop, further increasing elF4F and MYC translation. In addition, mTORC1 increases ChREBPβ, HIF1α, and p70S6K, which cumulatively increases lipogenesis, fatty acid uptake, glycolysis, and glucose uptake. The activation of this cascade is mediated by the PI3K/AKT pathway, which is normally inhibited by PTEN, as well as TSC1/2, which are produced by mTORC1 and provide a negative feedback loop to reduce mTORC1 activation. Mutations in PTEN, PI3K/AKT, or TSC1/2 will result in uncontrolled activation of mTORC1. mTORC2 increases SGK3 and AKT. AKT increases Cyclin D1 E, which plays a role in tumorigenesis and serves as a breast cancer biomarker, respectively.

**Figure 2 cells-12-02061-f002:**
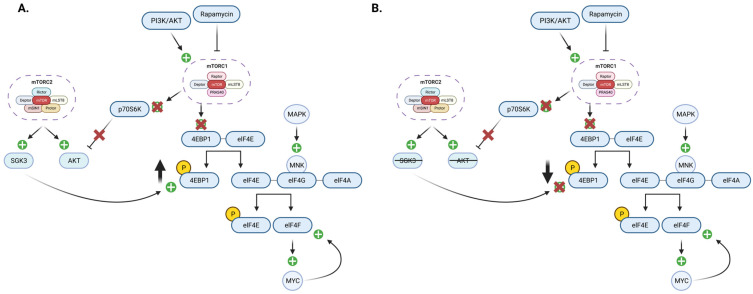
Rapamycin Resistance. (**A**) The pathway inhibition mediated by rapamycin is susceptible to resistance, reducing its efficacy. When rapamycin inhibits the mTORC1–p70S6K1 pathway, the negative feedback on AKT is removed, increasing AKT and resulting in only minor inhibition of 4EBP1 phosphorylation. SGK3 from mTORC2 further contributes to resistance by increasing phosphorylation of 4EBP1, bypassing rapamycin inhibition of mTORC1 to continue the pathway of elF4F and elF4E production. (**B**) Eliminating AKT and SGK3 restores the inhibition of mTORC1 by rapamycin, reducing phosphorylated 4EBP1 and subsequent elF4F and elF4E production.

## Data Availability

No new data were created or analyzed in this study. Data sharing is not applicable to this article.

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
