# Peer review of "The Role of Obesity in Breast Cancer Pathogenesis"

_cells, 2023, doi:10.3390/cells12162061_

Round 1
Reviewer 1 Report
Review comments on manuscript “The Role of Obesity in Breast Cancer Pathogenesis” by Glassman I., et al submitted to Cells.
In this review, authors discussed the role of obesity in breast cancer pathogenesis, which is an important area in cancer research. Obesity is known to contribute to breast cancer progression; however, how obesogenesis and obesity affect breast function, breast pathogenesis, and breast cancer development is an area of current debate. It is therefore important to keep scientific discussion and communications on the topic. This review is worth publishing, yet there are some issues need to be addressed.
the “introduction” section, authors discuss the association of obesity and type-2 DM with increased risk of breast cancer, the question is which one is more strongly associated with breast cancer. Please discuss.
In the “introduction” section, “Obesity and type-2 DM are considered states of chronic inflammation and oxidative stress” and “are associated with increased risk of various cancers, including breast cancer”. Does the breast cancer authors refer to here includes triple negative breast cancer (TNBC) or just hormone receptor positive (HR+) breast cancer? Authors need to corroborate more.
In the following paragraph, “BMI is inaccurate in assessing health” and “in premenopausal women, BMI-associated obesity is linked to protective effects”, are we able to infer that higher BMI in premenopausal women might be caused by higher muscle mass thus indicating a healthier state with lower cancer incidence? Please discuss.
In the section 2 “Obesity and Type II Diabetes”, authors describe the roles of NEFAs and TLR4 in linking obesity and type-2 DM. Do NEFAs and TLR4 are also further linked to breast cancer pathogenesis? If not, is it necessary to discuss them in this review?
“NFKB” in line 107 should it be “NFκB”?
In the section 3, “Obesity and Oxidative Stress”, it starts with “obesity correlates with over-activation of proinflammatory cytokines and immunologic cells”, however, authors only discus on the immunologic cells but not cytokines. Further, the crown-like structures (CLS) is an important feature of proinflammatory process in adipose tissue, it would be better to be also discussed in this part.
In the section 4, “Association between BMI and Breast Cancer”, authors describe the Women’s Health Initiative (WHI) clinical trials, the National Surgical Adjuvant Breast and Bowel Project Breast Cancer Prevention Trial (NSABP P-1) and others. Did these clinic trials further divide the breast cancer patients according to different subtypes? How was the result with TNBC?
If “there is concern about the utility of BMI in accurately assessing health”, was there any research using waist hip rate (WHR), body fat rate or other parameters to evaluate the relationship between obesity and breast cancer? It would be odd to discuss “the utility of BMI in accurately assessing health” between its role in postmenopausal and pre-menopausal women. Thus, it would be better to move this paragraph to line 187 before the paragraph “It has been established that increased BMI……”.
In the section 5 “Pathophysiology of Breast Cancer”, HER3 is not yet commonly used in breast cancer classification. Please clarify.
“This increased ROS production has been linked to the activation of several oncogenes and the inactivation of tumor suppressor genes, which are key factors in the development and progression of breast cancer”. It would be better to cite these genes here and give further discussion.
In line 285, why it was “At the hormonal level" when this paragraph is about hyperglycemia? Please clarify.
Is the pathophysiology of breast cancer similar or the same across different breast cancer subtypes? Would estrogen increase the TNBC incidence?
In the following sections, authors give a comprehensive discussion on “the role of the mTOR signaling pathway” and “the role of glutathione” in breast cancer. Although mTOR and glutathione were important mechanisms in obesity related breast cancer, other factors such as obese adipose tissue dysfunction, hypoxia microenvironment, the meta-inflammation should also be discussed or at least mentioned.
In line 449, “mitogen-activated protein kinase” is not used for the first time in the review and should be replaced by MAPK for short.
In line 454, “GSI synthesis”, is it a mistake for “GSH synthesis”?
In this review, “post-menopausal”, “postmenopausal”, “pre-menopausal” and “premenopausal”, are all randomly used, please make correction and be consistent.
It would be better to summarize the obesity-targeted therapies for breast cancer in preclinical and clinical trials other than leptin, if there is any.
Please double check and remove the spaces between words and symbols, such as row 29; 35; 95, etc.
It is acceptable.
Author Response
Reviewer #1
Review comments on manuscript “The Role of Obesity in Breast Cancer Pathogenesis” by Glassman I., et al submitted to Cells.
In this review, authors discussed the role of obesity in breast cancer pathogenesis, which is an important area in cancer research. Obesity is known to contribute to breast cancer progression; however, how obesogenesis and obesity affect breast function, breast pathogenesis, and breast cancer development is an area of current debate. It is therefore important to keep scientific discussion and communications on the topic. This review is worth publishing, yet there are some issues need to be addressed.
the “introduction” section, authors discuss the association of obesity and type-2 DM with increased risk of breast cancer, the question is which one is more strongly associated with breast cancer. Please discuss.
Author Response: Thank you for this feedback. We have included a discussion on which is more strongly associated.
In the “introduction” section, “Obesity and type-2 DM are considered states of chronic inflammation and oxidative stress” and “are associated with increased risk of various cancers, including breast cancer”. Does the breast cancer authors refer to here include triple negative breast cancer (TNBC) or just hormone receptor positive (HR+) breast cancer? Authors need to corroborate more.
Author Response: Thank you, this was certainly lacking from our original text. We have included more information on subtypes.
In the following paragraph, “BMI is inaccurate in assessing health” and “in premenopausal women, BMI-associated obesity is linked to protective effects”, are we able to infer that higher BMI in premenopausal women might be caused by higher muscle mass thus indicating a healthier state with lower cancer incidence? Please discuss.
Author Response: More content has been added regarding the association between BMI in premenopausal women and cancer risk, as well as an exploration on confounding variables such as muscle mass. Thank you for this great point.
In the section 2 “Obesity and Type II Diabetes”, authors describe the roles of NEFAs and TLR4 in linking obesity and type-2 DM. Do NEFAs and TLR4 are also further linked to breast cancer pathogenesis? If not, is it necessary to discuss them in this review?
Author Response: Information regarding NEFAs and TLR4 as they relate to breast cancer pathogenesis has been added, thank you.
“NFKB” in line 107 should it be “NFκB”?
Author Response: Yes, thank you for catching this.
In the section 3, “Obesity and Oxidative Stress”, it starts with “obesity correlates with over-activation of proinflammatory cytokines and immunologic cells”, however, authors only discus on the immunologic cells but not cytokines. Further, the crown-like structures (CLS) is an important feature of proinflammatory process in adipose tissue, it would be better to be also discussed in this part.
Author Response: Thank you for this point. More information regarding specific cytokines has been added throughout the paper, as well as information regarding CLS as it pertains. We appreciate this recommendation.
In the section 4, “Association between BMI and Breast Cancer”, authors describe the Women’s Health Initiative (WHI) clinical trials, the National Surgical Adjuvant Breast and Bowel Project Breast Cancer Prevention Trial (NSABP P-1) and others. Did these clinic trials further divide the breast cancer patients according to different subtypes? How was the result with TNBC?
Author Response: NSABP P-1 and STAR did happen to mention subtyping and this has been added to the paper, thank you.
If “there is concern about the utility of BMI in accurately assessing health”, was there any research using waist hip rate (WHR), body fat rate or other parameters to evaluate the relationship between obesity and breast cancer? It would be odd to discuss “the utility of BMI in accurately assessing health” between its role in postmenopausal and pre-menopausal women. Thus, it would be better to move this paragraph to line 187 before the paragraph “It has been established that increased BMI……”.
Author Response: Thank you for this. We have moved that paragraph to where you suggested and added a study regarding waist circumference.
In the section 5 “Pathophysiology of Breast Cancer”, HER3 is not yet commonly used in breast cancer classification. Please clarify. –
Author Response: Thank you. Yes, this is correct, and has been removed.
“This increased ROS production has been linked to the activation of several oncogenes and the inactivation of tumor suppressor genes, which are key factors in the development and progression of breast cancer”. It would be better to cite these genes here and give further discussion.
Author Response: Thank you for the suggestion, this has been added.
In line 285, why it was “At the hormonal level" when this paragraph is about hyperglycemia? Please clarify.
Author Response: Correct, this has been changed to metabolic, thank you.
Is the pathophysiology of breast cancer similar or the same across different breast cancer subtypes? Would estrogen increase the TNBC incidence?
Author Response: Thank you for this. We have added more information regarding TNBC and estrogen to differentiate from the other subtypes.
In the following sections, authors give a comprehensive discussion on “the role of the mTOR signaling pathway” and “the role of glutathione” in breast cancer. Although mTOR and glutathione were important mechanisms in obesity related breast cancer, other factors such as obese adipose tissue dysfunction, hypoxia microenvironment, the meta-inflammation should also be discussed or at least mentioned.
Author Response: This is a great suggestion. We have added more information regarding these subjects throughout the paper.
In line 449, “mitogen-activated protein kinase” is not used for the first time in the review and should be replaced by MAPK for short.
Author Response: This has been corrected, thank you.
In line 454, “GSI synthesis”, is it a mistake for “GSH synthesis”?
Author Response: This has been corrected, thank you.
In this review, “post-menopausal”, “postmenopausal”, “pre-menopausal” and “premenopausal”, are all randomly used, please make correction and be consistent.
Author Response: This has been corrected, thank you.
It would be better to summarize the obesity-targeted therapies for breast cancer in preclinical and clinical trials other than leptin, if there is any.
Author Response: Thank you for this suggestion, we have added obesity-targeted therapies to that section and created a new section on immunotherapies.
Please double check and remove the spaces between words and symbols, such as row 29; 35; 95, etc.
Author Response: Corrected, thank you.
Thank you for all of your feedback, we feel it has greatly improved the quality of our paper.

Reviewer 2 Report
This review describes the role of obesity in developing breast cancer. However, it is very general and matches several other reviews in the topic, and missing many aspects of obesity. There are studies that describe obesity as protective in premenopausal women which is not mentioned here. Additionally, different obesity profile could be mentioned. Central obesity for eg is linked to BC development, and %body fat is also reported.
ALso focusing on mTOR pathway is a bit out of scope, there are many other obesity linked pathways that could be discussed, inflammation induced obesity for eg.
minor english edits
Author Response
Reviewer #2
This review describes the role of obesity in developing breast cancer. However, it is very general and matches several other reviews in the topic and missing many aspects of obesity. There are studies that describe obesity as protective in premenopausal women which is not mentioned here. Additionally, different obesity profile could be mentioned. Central obesity for eg is linked to BC development, and %body fat is also reported.
Also focusing on mTOR pathway is a bit out of scope, there are many other obesity linked pathways that could be discussed, inflammation induced obesity for eg.
Author Response: Thank you for your feedback. We have added more in-depth discussion exploring other therapies, classification of breast cancer and discussions on obesity and pathophysiology. We appreciate your insight and believe it has contributed greatly to the quality of our paper.

Round 2
Reviewer 1 Report
Review comments on the revised manuscript “The Role of Obesity in Breast Cancer Pathogenesis” by Glassman I., et al submitted to Cells.
In the revised manuscript, authors have addressed our major questions and suggestions raised in the first review. This is only one minor question: In line 53, molecular subtypes of breast cancer are determined on the expression level of hormone-receptor, (including estrogen-receptor and progesterone-receptor), human epidermal growth factor 2 (HER2) and Ki-67. The difference between subtype Luminal A and Luminal B is not merely the HER2 state, ki-67 level is more important. If addressed, we believe that this review is worth publishing.
Minor check
Author Response
Reviewer 1 Comments: In the revised manuscript, authors have addressed our major questions and suggestions raised in the first review. This is only one minor question: In line 53, molecular subtypes of breast cancer are determined on the expression level of hormone-receptor, (including estrogen-receptor and progesterone-receptor), human epidermal growth factor 2 (HER2) and Ki-67. The difference between subtype Luminal A and Luminal B is not merely the HER2 state, ki-67 level is more important. If addressed, we believe that this review is worth publishing.
Author Response: Thank you for your feedback. We have amended line 53 to include Ki-67 in the differentiation of subtypes. We also changed the broad hormone receptor (HR) term to include both estrogen and progesterone specific receptors (ER & PR) to further define the subtypes.